# LncRNAs and Immunity: Coding the Immune System with Noncoding Oligonucleotides

**DOI:** 10.3390/ijms22041741

**Published:** 2021-02-09

**Authors:** Marco Bocchetti, Marianna Scrima, Federica Melisi, Amalia Luce, Rossella Sperlongano, Michele Caraglia, Silvia Zappavigna, Alessia Maria Cossu

**Affiliations:** 1Laboratory of Molecular Oncology and Precision Medicine, Biogem Scarl, Institute of Genetic Research, 83031 Ariano Irpino, Italy; marianna.scrima@biogem.it (M.S.); federica.melisi@biogem.it (F.M.); michele.caraglia@unicampania.it (M.C.); 2Department of Precision Medicine, University of Campania “Luigi Vanvitelli”, 80138 Naples, Italy; amalia.luce@unicampania.it (A.L.); silvia.zappavigna@unicampania.it (S.Z.); 3School of Science and Technology, Nottingham Trent University, Clifton Lane, Nottingham NG11 8NS, UK; 4Department of Experimental Medicine, University of Campania “Luigi Vanvitelli”, 80138 Naples, Italy; rossella.sperlongano@unicampania.it

**Keywords:** long noncoding RNA, immunity, inflammation, autoimmune diseases, transcript regulation

## Abstract

Long noncoding RNAs (lncRNAs) represent key regulators of gene transcription during the inflammatory response. Recent findings showed lncRNAs to be dysregulated in human diseases, such as inflammatory bowel disease, diabetes, allergies, asthma, and cancer. These noncoding RNAs are crucial for immune mechanism, as they are involved in differentiation, cell migration and in the production of inflammatory mediators through regulating protein–protein interactions or their ability to assemble with RNA and DNA. The last interaction can occur in cis or trans and is responsible for all the possible lncRNAs biological effects. Our proposal is to provide an overview on lncRNAs roles and functions related to immunity and immune mediated diseases, since these elucidations could be beneficial to untangle the complex bond between them.

## 1. Introduction

Long-chain noncoding RNAs (lncRNAs) are a new class of transcripts found to be commonly transcribed in the genome, generally characterized by nonprotein transcript, regulating many biological aspects related to human diseases. LncRNAs are gaining importance for cancer as they directly interact with the cell cycle, proliferation pathways and microbiome balance. They are involved in the activation of the innate immune response and T cell development, differentiation, and activation [1]. LncRNAs are defined as endogenous cellular RNA molecules presenting more than 200 nucleotides in length that lack an open reading frame (ORF) of significant length (less than 100 amino acids) and contain one or more exons [2]. Over the past decade, over 18,000 transcripts have been recognized and annotated as lncRNA in mammalian transcriptomes, thanks to rapid advances in high-throughput genetic sequencing technology [3,4]. The sequencing data accumulation allowed the identification of an increasing lncRNAs number, since some of them were previously mistakenly identified as protein coding genes, and now are being correctly annotated [5]. The high sequence conservation of lncRNA promoters is even higher than the protein coding genes, suggesting that regulation of lncRNA expression is extremely important [6]. On the other hand, various studies revealed the importance of lncRNAs as they are involved in a series of functionally distinct biological and physiological processes including chromatin remodeling, gene transcription, RNA junction, protein transport [7]. Moreover, lncRNAs are directly linked to human diseases including various types of cancer, such as head and neck cancers [8,9], Alzheimer’s disease [10], and coronary heart disease [11]. LncRNAs are known to play important roles in other biological pathways, including modulation of the innate immune response; the exact function and operative mechanism of those lncRNAs are not yet completely characterized [12,13]. Recently, different studies demonstrate the regulations related to lncRNA in different types of immune coding and noncoding genes and their role in autoimmune diseases [14]. Furthermore, lncRNAs act as key regulators of inflammatory gene expression by interacting with signal-dependent activation of transcription factors, transcriptional coregulators and chromatin-modifying factors [15]. To date, the exact mechanisms of lncRNAs functions in autoimmunity are not completely clear. Autoimmune diseases represent the third most common diseases after cardiovascular disease and cancer. The onset of these particular immunity-related diseases is very complicated and is caused by the interaction and balance between genetic and environmental factors. LncRNAs are becoming a new hope in the autoimmune disease paradigm, considering their great effects on the adaptive and innate immune system. Several investigations have indicated a strong correlation between autoimmune disease and the lncRNAs level balance. For example, overexpression of lncRNA DQ786243 was found in patients with Crohn’s disease [16], overexpression of lncRNA HIVEP2 in patients with systemic lupus erythematosus (SLE) [17], lncRNA SAS-ZFAT plays a role in autoimmune thyroid disease [18]. The lncRNAs role is a growing field in disease and immune system regulation and a better knowledge of lncRNAs is required, since they possibly have a diagnostic-prognostic value in the treatment of inflammatory diseases. Here, we are going to review the lncRNAs functions associated to adaptive and innate immunity and to the autoimmune diseases, as well.

## 2. Inflammatory Response

Inflammation is a highly preserved multifactorial and complex process, that affects physiological processes and diseases [19]. This process represents an essential response to harmful stimuli, such as pathogens, damaged cells, toxic compounds or irradiation causing tissue damage, maintaining tissue homeostasis [20]. In response to tissue injuries, in the body, a chemical signaling cascade starts responsible for the leukocyte activation and chemotaxis from the general circulation to the damage sites in order to release cytokines that induce inflammatory responses [21]. In detail, the first step of innate immune response involves the recognition of infection or damage through the detection of molecular patterns associated with pathogens (PAMPs), invaders and molecular patterns associated with danger (DAMPs), and recognition of receptors of germline-encoded schemes (PRR) (Figure 1). Bacterial pathogens are detected by innate immunity system receptors, such as Toll-like receptors (TLRs), which are expressed on tissue-resident macrophages and able to induce, after the binding with their ligands, the production of inflammatory cytokines (e.g., TNF-α(Tumor necrosis factor), IL-1, IL-6(Interleukins) and chemokines (e.g., CCL2 and CXCL8, C-X-C Motif Chemokine Ligand 8), as well as prostaglandins. In particular, neutrophils, dendritic cells, and macrophages are recruited from the circulation in response to damage by Toll-like receptors (TLR), type C lectin receptors (CLR), inducible retinoic acid (RIG)-I-gene receptors like (RLR) and NOD-like receptors (NLR) [22]. TLRs are a family of highly conserved mammalian PRRs. TLRs participate in the activation of inflammatory response and represent the main sensors evolutionarily preserved as pathogen-associated models, involved in the transmission of signals from the plasma membrane through a multistep cascade of reactive transcription factors.

Unlike adaptive immunity, the innate immune system lacks the ability to distinguish between different strains of pathogens and whether these strains are virulent (harmful to the host). Once ligand recognition occurs, TLRs activate NF-kB (nuclear factor kappa-light-chain-enhancer of activated B cells) signaling pathway. NF-kB is located in the cytosol bound to an IkB (inhibitor of kB) inhibitory protein (IkBα) in its blocked state. 

PAMPs and DAMPs signal regulate either MyD88-dependent or TRIF-dependent pathways. Signaling through TLRs activates intracellular signaling cascades that lead to nuclear translocation of AP-1 and NF-κB or IRF3, which regulate the inflammatory response. NF-κB is triggered by TLRs and inflammatory cytokines, such as TNF and IL-1, leading to activation of RelA/p50 complexes that regulate expression of inflammatory cytokines. NF-κB signaling requires IKK subunits for pathway activation through IκB phosphorylation. MAPK mediates intracellular signaling initiated by extracellular stimuli, such as stress and cytokines. In particular DAMP signaling is responsible for inflammasome assembly, caspase-1 activation, and secretion of IL-1β and IL-18.

Its activation starts with IKK stimulation and consequent IkB phosphorylation. Phosphorylated IkB is then ubiquitinated and degraded in the proteasome. In this way, NF-κB becomes available and is translocated to the nucleus where it binds to specific DNA sequences called response elements (RE). The DNA/NF-κB complex engages other proteins such as coactivators and RNA polymerase transcribing DNA into mRNA, finally exported to the cytosol and translated into proteins leading to the production of proinflammatory cytokines. NF-κB, at the same time, also activates the transcription of IkB inhibitory subunit, thus generating a negative feedback loop. Moreover, inflammatory process is characterized by vascular permeability changes, leukocyte recruitment and accumulation, and inflammatory mediator release [23]. STAT3 (Signal transducer and activator of transcription) is an important inflammation mediator and also opposes to NF-κB and STAT1 dependent immune responses promoting antitumor immunity. STAT3 can also promote pro-oncogenic inflammatory pathways, including nuclear factor-IkB (NF-kB) and interleukin-6 (IL-6)-GP130-Janus kinase (JAK) pathways. Interleukin-6 (IL-6) represents STAT family proteins activator together with Janus kinase (JAK) tyrosine kinases (JAK1, JAK2, JAK3, and TYK2). MAPK signaling can induce the release of proinflammatory mediators, such as cytokines and chemokines, and activate anti-inflammatory pathways, following Toll-like receptors (TLRs) activation. In particular, p38α can activate protein kinase 2 which stimulates the production of tumor necrosis factor (TNF). TLRs recognize IL-1R (IRAK) and TNFR-associated factor 6 (TRAF6), and this complex activates TGFβ-activated kinase 1 (TAK1), a MAPK kinase (MAP3K) upstream of p38α, and Jun N-terminal kinase (JNK). TAK1 can also activate the IκB kinase complex (IKK), leading to NF- κB activation and the tumor progression locus MAP3K 2 (TPL2), which is upstream regulated by extracellular kinase 1 (ERK1) and ERK2 [24]. The inducible expression of proinflammatory cytokines leads to the third stage of the inflammatory cascade. Together with the chemokines (attractants) and various costimulatory molecules, these soluble proteins facilitate the recruitment of effector cells such as monocytes and neutrophils at the injured site. Neutrophils release harmful chemicals (ROS, Reactive Oxygen Species and RSN, Reactive Nitrogen Species) for pathogens through a process called degranulation with the aim of destroying pathogens [24]. After the first few hours of inflammation (acute inflammation), macrophages are recruited and are responsible for the release of proinflammatory prostaglandins and leukotrienes, and then lipoxins. Those exert negative feedback mechanism on neutrophils recruitment and favor the enhanced infiltration of monocytes important for wound healing [25]. Unlike innate immune response, adaptive response is highly specific for a given pathogen responsible for its activation. Occasionally, the system cannot distinguish between pathogen and host-derived antigens engaging autoimmune diseases. Lymphocytes are responsible for both induction and expression of adaptive immunity. There are two main classes of lymphocytes, consistently produced by precursor stem secretion of cytokines and adaptive immune system antibody producing cells [26]. T lymphocytes are adaptive immunity cells responsible for protecting the host against infections. During maturation, T lymphocytes produce and expose the antigen receptor called T-cell receptor (TCR) on cell membrane. Each receptor is able to recognize the antigen based on its chemical structure; in particular, TCR recognizes peptides, only if they are presented by class I or class II HLA (Human Leukocyte Antigens) molecules on plasma membrane of APC cells. Antigen presenting cells (APCs) are the first players involved in our immune system, being part of innate immunity. Often dendritic cells and macrophages internalize and disrupt pathogens; then they expose part of them on their outer membrane, bound to MHC class I or II complex. These presented small digested peptides, together with costimulatory T cell molecules, are required to activate the more specific adaptive immunity. T cells membrane glycoproteins are also called coreceptors: CD4 and CD8 (Cluster of differentiation) play a role in the recognition and activation of T cells by binding to their respective ligands of the class II and class I histocompatibility complex (MHC) on the antigen-presenting cell (APC), thus increasing the adhesion between T cells and APCs. Both coreceptors bind to HLA molecules in a different site, but CD4 is specific for class II HLA while CD8 can only bind class I HLA molecules. CD4 and CD8 T lymphocytes circulate continuously in the blood, lymphatic stream, and through the secondary lymphatic organs in search of microbial antigens.

NF-kB, activator protein 1, STAT-JAK and MAPK represent the transcription factors and pathways that regulate inflammatory response [27,28,29].

Uncontrolled acute inflammation may become chronic, contributing to a variety of chronic inflammatory diseases, like metabolic disorders [30]. Chronic inflammatory diseases share common characteristics, including elevated circulating levels of cytokines TNF-α and IL-6 under basal or resting conditions.

## 3. LncRNAs Molecular Features

LncRNAs are RNA molecules with more than 200 nucleotides abundant in genome, some RNA molecules are bifunctional with coding and noncoding capabilities [31]. GENECODE project (www.genegodegenes.org, accessed on 12 February 2019) defined lncRNAs as the largest class of nonprotein-coding RNAs, with 16,000 identified in humans and 9000 in mice [Gencode. GENCODE. Statistics about the Current GENCODE Release (version 29). 2019. Available online: https://www.gencodegenes.org/human/stats.html (accessed on 12 February 2019)].

LncRNAs originated more than 300 million years ago and can be found from organisms ranging from Xenopus and chicken to man [32]. Many lncRNAs are lowly expressed [33], and this is why they might be defined “transcriptional noise” posing a challenge in terms of discovery of lncRNAs. Therefore, lncRNA classification is based on their genomic localization and orientation relative to protein-coding genes, and can be either long intergenic noncoding RNAs (lincRNAs; between protein-coding genes) [34], intronic (within introns of protein-coding genes), antisense to protein-coding genes (natural antisense transcripts, NATs) [35], or transcribed from divergent bidirectional promoters [36]. LncRNAs expression is specific for cells and tissues, confirmed by transcriptome studies [37,38,39]. Unlike the protein-coding genes, lncRNAs have fewer but longer exons and are generally poorly evolutionarily conserved; this makes it difficult to search for related domains and compare their biological meaning between species, but they have highly conserved genomic positions [40]. However, today we know some lncRNAs characterized by high conservation of sequences and structure levels: MALAT1 (transcription 1 of lung adenocarcinoma associated with metastasis 1) and NEAT1 (transcription abundantly enriched by nuclear 1) [41,42].

### LncRNAs Main Biological Mechanisms

LncRNA mediate chromatin remodeling, usually exerted locally (in cis), but it can also occur in trans. In general, lncRNAs regulate gene expression through interactions with DNA, RNA, or proteins. There are other important lncRNA-mediated mechanisms: translation control, epigenetic silencing, cell cycle regulation, and apoptosis. The subcellular localization of lncRNA is very important, since it provides useful information including the modal investments of the long noncoding action. About 30% of lncRNAs are known to be localized in the nucleus. There, after the transcription of target genes and through the interaction with transcription factors, they are able to form chromatin-modifying complexes, or interact with heterogeneous ribonucleoprotein complexes (hnRNP), a class of nuclear proteins involved in precursor mRNA processing. In cytosol, however, lncRNAs have been shown to control mRNA stability in a positive way: in the immune system an example of this mechanism is exerted by iNOS-AS, an antisense RNA, which is encoded on the antisense strand corresponding to the 3′UTR of the iNOS mRNA [43]. In contrast, cytoplasmic antisense lncRNAs have often been found to have the opposite action; their direct complementarity with mRNAs target lead to gene silencing associated with Dicer [34,44,45]. Moreover, lncRNAs have a target-mimetic, sponge/bait function on microRNAs [46]; they act as miRNA sponges, reducing their regulatory effect on mRNAs. Recently, a computational method called LMSM was identified to predict miRNA targets and their possible interaction with lncRNAs in breast cancer [47].

## 4. LncRNAs and Regulation of Immunogenic Expression

LncRNAs play an important role in the regulation of immune system, but due to its complexity and diversity lncRNAs regulation understanding remains challenging. Several studies have highlighted the presence of lncRNAs in immune cells which are responsible for development, differentiation, and their activation [6,48]. Regulatory functions of many immune-related lncRNAs are above all RNA/protein binding or RNA/DNA base pairing processes [49]. Most of them partially overlap or are close to the 3′ (downstream) or 5′ (upstream) end of the protein-coding genes, playing key roles in regulating immune response mechanisms [50]. Usually, lncRNAs and mRNAs use a common promoter region to produce bidirectional transcription. The transcription of immune-related lncRNAs occurs in the antisense direction. Surprisingly, antisense lncRNAs often have no or have only a partial overlap with protein-coding genes [51]. To date, some lncRNAs are known to control gene expression involved immune response mechanism either in cis or in trans. In this context, HOTAIR is a lncRNA located in intergenic regions, acting in trans [52], while other lncRNAs regulate their genes encoding adjacent proteins in cis or in transactive ways, like PACER [53], lnc-IL7R [54], THRIL [55], and lincR-Ccr2-50AS [50]. It is important to underline how immune-related lncRNAs are direct or indirect targets on specific transcription factors responsible for inflammatory mediator production [48,56]. Not surprisingly, many lncRNAs are identified and involved in different pathways, such as NF-kB, arachidonic acid, MAPK, and JAK/STAT signaling regulating inflammation and immune response [57].

## 5. LncRNAs and the Innate Immune Response

An increasing number of lncRNAs have been found to play roles in innate immune response. Therefore, we tried to focus our attention on widely studied and recently identified lncRNAs, describing them for the different functions and modes of action (Table 1).

*ANRIL*. ANRIL (Antisense Noncoding RNA in the INK4 Locus) is a long noncoding RNA transcribed from chromosome 9p21. A lot of studies showed the different strategies of ANRIL activations. ANRIL in human umbilical vein endothelial cells (HUVECs) is activated by NF-kB and is important for the expression of a subset of proinflammatory genes. Moreover, a recent work explored the regulatory mechanism of ANRIL on IL6/8; Yin Yang 1 (YY1), an ANRIL binding transcriptional factor is required for IL6/8 expression under TNF-α treatment. In particular, ANRIL and YY1 cooperate during the inflammatory gene regulation. This study evidenced YY1 to be enriched at promoter loci of *IL6/8* and ANRIL silencing impaired this phenomenon. *ANRIL* was found to be a novel member of TNF-α-NF-kB pathway involved in the inflammatory factor activation under pathological stresses [58].

*AS-IL1α.* AS-IL1α is a natural antisense lncRNA partially complementary to *IL-1α*. This natural antisense can be activated by cytokines-mediated inflammatory response. Infection mediated by *Listeria monocytogenes* or stimulation with TLR ligands (Pam3CSK4, LPS, polinosinic-polycytidyl acid) induce AS-IL1α expression in macrophages. AS-IL1α is located in the nucleus and does not alter the stability of *IL-1α* mRNA. Chan et al. have demonstrated in IL-1α deficient mice macrophage cells, that LPS stimulation induces the expression of AS-IL1α, but not of *IL-1α*, indicating that the transcription of AS-IL1α is regulated independently from *IL-1α*. Consequently, disruption of AS-IL1α function can limit IL-1α transcription and reduce the harmful effects of excessive IL-1α levels during infection and inflammatory disease. AS-IL1α is an important regulator of *IL-1α* transcription during the innate immune response [59].

*FIRRE.* This high conserved lncRNA is a nuclear-organization factor located on the X chromosome and expresses its function in the nucleus interacting with heterogeneous nuclear ribonucleoproteins (hnRNPU) [84]. FIRRE transcription is controlled by NF-kB signaling in macrophages and intestinal epithelial cells and is involved in the regulation of many inflammatory genes at the post transcriptional level in response to LPS stimulation. In particular, FIRRE physically interacts with heterogeneous nuclear ribonucleoproteins U, regulates the stability of mRNAs of selected inflammatory genes by targeting the AU-rich elements of their mRNAs in cells following LPS stimulation. FIRRE knockdown in nonstimulated and LSP-stimulated SW480 cell lines decreases mRNA levels of *IL-12p40*, *IL-1b*, *TNF-α*, *G-CSF*, and *VCAM-1*, along with increased expression of IL-6. On the other hand, FIRRE overexpression increases *IL-12p40*, *VCAM1*, *IL-1A*, *ICSF1* mRNA levels and decreases *CXCL10*, *IL-8*, *IL-1B*, *RANTES*, and IL-6. Moreover, FIRRE and hnRNPU show a synergistic action in increasing the VCAM-1 mediated chemotaxis on Jurkat cells [60].

*H19.* H19 plays a crucial role in maintaining adult hematopoietic stem cell quiescence. Venkatraman et al. investigated on mice models and on single CD34−Flk2−LSK cells the maternally specific upstream region of H19, called *H19-DMR*. This region activates the *Igf2–Igf1* pathway by the translocation of the phosphorylated FoxO (Forkhead box O3) to the cytoplasm. This leads enhanced *Igf2* expression and *Igfr1* translation, playing a significant role in the increased activation and proliferation of hematopoietic stem cell (HSCs) [61].

*IL-1b -RBT46*. IL-1b-RBT46 (bidirectional transcription regions IL-1b 46) upregulation is activated in monocytes by TLR-4. IL-1b-RBT46 shares a promoter with the coding gene *IL-1b*. As reported, the knockdown of nuclear-localized, NF-κB-regulated, eRNAs (*IL1β-eRNA*), and RBT (*IL1β-RBT46*) surrounding the *IL-1β* locus, attenuates LPS-induced messenger RNA transcription and release of the proinflammatory mediators: IL1β and CXCL8 [85].

*IL7-AS*. In a recent report, IL7-AS was evaluated and characterized in terms of expressions and regulation of immunity using next generation sequencing. During this analysis, the syntenic antisense lncRNA *IL7-AS*, is induced in different cell types: human A549 lung epithelial cells, LPS-stimulated human monocytic THP-1 cells and LPS-stimulated mouse RAW 264.7 macrophages. IL7-AS was able to regulate the production of IL-6, an important proinflammatory mediator in both human and mouse cells. For IL7-AS role evaluation during the innate response, was used AS locked nucleic acid (LNA) to knockdown the lncRNA in human and in mouse models. The results suggested that any potential biological actions of IL7-AS might be mediated in trans. IL7-AS acts as a positive regulator of IL1β-induced inflammatory response in human A549 epithelial cell, but as a negative regulator of IL1β-stimulated IL6 release in human chondrocytes, in LPS-stimulated human THP-1 monocytes and mouse RAW 264.7 macrophage [63].

*Lethe.* The long noncoding Lethe is implicated in NF-kB activity. *Lethe*, is a pseudogene, expressed in mouse embryonic fibroblasts. This lncRNA, interacts with RelA-RelA homodimers influencing the NF-κB response elements. In particular, it acts as a part of a negative feedback loop, after NF-κB activation by proinflammatory cytokines [64]. Dysregulation of Lethe expression has been hypothesized to play a role in hyperglycemia-induced ROS production. In fact, the RAW264.7 macrophages, treated with different glucose concentrations to evaluate over time the *NOX2* gene expression variation and the ROS production, showed that high glucose concentrations significantly induced ROS production and *NOX2* gene expression in treated cells, significantly reducing Lethe expression. While, its overexpression in RAW cells eliminated ROS production induced by elevated glucose conditions, also attenuating the upregulation of *NOX2* expression. Furthermore, Lethe overexpression in RAW glucose-treated cells significantly reduced p65-NF-kB translocation in the nucleus, causing a reduction in *NOX2* expression and ROS production, demonstrating that Lethe lncRNA is involved ROS production regulation in macrophages through NF-kB-related *NOX2* gene expression modulation [65].

*LincRNA-COX2*. One of the most relevant lncRNA, is LincRNA-COX2. It is located in the intergenic region of a protein coding gene and highly induced in innate immunity cells after the activation of TLRs. COX2 proteins are involved in arachidonic acid metabolism, generating PGEs. LincRNA-COX2 induction, mediated by NF-kB, is involved in the regulation of more than 700 genes. It has been reported that this lincRNA, can suppress TNFα pathway and the consequent transcription of *IL-12b* gene via epigenetic modifications. Moreover, LincRNA-COX2 appears to be an early product of NF-kB pathway, with expression pike around 60 min after LPS stimulation. LincRNA-COX2 can mediate the late-primary innate response through chromatin remodeling when it is associated with SWI/SNF complex in macrophages and microglia acting on RelA and p50 subunits. In particular *CCL-5*, *CXC-L10*, *Peli1*, *Traf1*, *SAA3*, and *IFNB-1* gene expression share a positive correlation with LincRNA-COX2 expression [15]. The role of LincRNA-COX2 was investigated by generating two murine models and multiple macrophage cell lines, which strengthened the correlation: LincRNA-COX2 inhibits ISGs expression and enhances proinflammatory gene expression. In addition, the CRISPRi tool was used to elucidate LincRNA-COX2 functions, demonstrating the eRNA mechanism to control the expression of the neighbor gene *Ptgs2* (Cox2) in vivo. Finally, it was determined whether LincRNA-COX2 could affect gene expression in the periphery following LPS challenge in vivo. LincRNA-COX2 works in *trans* controlling different innate immune genes, increasing CCL5 and IP10 (CXC-L10) levels and reducing IL-5 and IL-17 in mice serum [66].

*Lnc-DC*. Lnc-DC is expressed in conventional human dendritic cells (DC), binds directly to STAT3 in the cytoplasm, promoting the STAT3 phosphorylation on tyrosine-705. The study by Wang et al. showed that the DC-DC knockdown altered the differentiation of DC from human monocytes in vitro and mouse bone marrow cells in vivo and reduced the DCs ability to stimulate T cell action. Lnc-DC mediates these effects by activating the transcription factor of STAT3 [67].

*LincRNA-EPS*. This long intergenic noncoding RNA is expressed in macrophages and dendritic cells, originally known as erythrocyte differentiation regulator [86], was found remarkably downregulated in TLR2-activated mouse bone marrow macrophages compared to resting macrophage. The lincRNA suppression occurs following the stimulation of all three TLR ligands and is time-dependent. The most significant upregulation involves cytokine and chemokine genes such as *CXC110*, *CXC-L9*, *TNFsf-10*, *TNFsf-8*, *IL-27*, IFN-stimulated genes *IFIT2*, *Rsad2* and, *Oas11*. To deeply understand lincRNA-EPS mechanism, it was upregulated in macrophages and this caused severe impairment in the expression of cytokines, chemokines, and antiviral ISGs. After localization studies, it was found associated to the chromatin in resting macrophages acting on IRGs expression by binding their regulatory regions and maintaining chromatin in an epigenetically repressed state, while no activating signals were given. In vivo models were generated by deleting the whole 4 kb lincRNA-EPS locus. Animals did not show any abnormalities in hematocrit, growth, and health in general. Following LPS stimulation, knockout mice showed an enhanced inflammatory response overall and many IRG were upregulated compared to wild type mice at different time-points. Furthermore, the augmented levels of CXC-L10, CCL-5, and IL-6 were observed in lincRNA-EPS deficient mice peritoneal fluid, serum, and spleen. The absence of the control on the IGS exerted by the lincRNA made the mutated mice more susceptible to endotoxic-shock, with a death rate increased by 50% compared to wild type mice after sublethal LPS injection. Even after a sublethal LPS injection, the lincRNA -/- mice mortality was 80% while no wild type mice died after 5 days post-injection [62].

*LncHSC-1/2.* LncHSC-1 and LncHSC-2 exhibit several interesting features and their promoter regions are bound by multiple TFs in hematopoietic progenitor cells. LncHSC-1 is more HSC specific, and LncHSC-2 is expressed in HSC and also different progenitors, but not terminally differentiated cells. LncHSC-1 and LncHSC-2 are located into the nucleus, and differentially expressed between WT and *Dnmt3a* KO HSCs. KD of *LncHSC1/2* revealed that LncHSC-1 is involved in myeloid differentiation, and LncHSC-2 is involved in HSC self-renewal and T cell differentiation [68].

*Lnc-IL7R.* Lnc-IL7R (interleukin-7 receptor α-subunit) heavily contributes in inflammatory response regulation. Lnc-IL7R (interleukin-7 receptor α-subunit) is rapidly induced by LPS stimulation of human monocytic THP-1 cells and human peripheral blood mononuclear cells (PBMNC). Lnc-IL7R has also been reported to negatively regulate the expressions of *IL-6*, *IL-7R*, *IL-8*, *VCAM-1,* and *E-selectin*. Lnc-IL7R is induced by ligand engagement of TLR2 or TLR4, and the activation of NF-kB, while the stimulation of TLR3 primarily produces type I interferon responses. Reduced levels of H3K27me3, a hallmark of transcriptionally repressed chromatin, have been found in regions promoting the *E-selectin* and *VCAM-1* genes in LPS-stimulated cells. Lnc-IL7R appears to regulate H3K27me3 since, its downregulation further reduces H3K27me3 levels to the promoters of *E-selectin* and *VCAM-1*, a discovery consistent with the elevated expression of these genes in Lnc-IL7R knockdown cells. However, no significant changes were found in H3K29me3 in the *IL-8* promoter in Lnc-IL7R knockdown cells, suggesting that the increased expression of IL-8 in these cells is mediated by other mechanisms. Huachun et al. have revealed that Lnc-IL7R knockdown decreased the trimethylation of histone H3K27 at promoters of inflammatory mediators, suggesting that Lnc-IL7R epigenetically regulates inflammatory responses. [54].

*LncITPRIP-1*. This lncRNA, located on chromosome 10 adjacent to the coding gene *DANGER*, was found highly upregulated after in vitro INF-α stimulation on human hepatocytes. INFs I and III are the major players against viral infections [87]. Host antiviral response starts with IFNs signal from the JAK/STAT pathway interferon-stimulated genes (IGSs) activation [88,89]. LncITPRIP-1 levels were evaluated in several in vitro models like Huh7, Huh7.5, huh7.5.1-MAVS, and HLCZ01 after HCV, HSV, SeV, and HSV infection. This lncRNA was upregulated in virus in an amount/time-dependent manner. Moreover, cell lines expressing higher LncITPRIP-1 levels showed higher resistance to the virus, lower replication and enhanced INF production, as well. In particular, LncITPRIP-1 anti-HCV capability was tested by knocking in and out this lncRNA from Huh7.5, FL-neo, and HLCZ01 cells. All the results were in agreement and showed that ectopic expression of the lncRNA was attenuating and impairing virus replication, while its loss or downregulation was mildly enhancing viral replication and also increasing HCV RNA levels and proteins. As expected, since it has been correlated with INF pathway, the lncRNA upregulation in HLCZ01 increases also INF-β, IL-28A, IL29 production and upregulated ISG12a, ISG56, ISG60, and IRF3. Probably, this action was due to the MDA5-dependent IFN pathway activation since MDA5 and MAVS protein levels were found upregulated in LncITPRIP-1 overexpressing cells. MDA5, a cytosolic enzyme receptive to viral infection [69] was confirmed to interact specifically with *lncTPRIP-1* by RIP assay and is required to exploit its HCV inhibition [90].

*Lnc-Lsm3b*. Lnc-Lsm3b competes with the viral RNA in the binding of the RIG-I monomers and inactivates the innate RIG-I function in the advanced phase of the innate response. RIG-I is a key sensor and is involved in antiviral type I IFNs production after the recognition of “non-auto” viral RNAs. Lnc-Lsm3b probably limits the conformational shift of the RIG-I protein, and thus interrupts the production of type I IFN. In conclusion, it acts as a regulator of the innate immune response and in maintaining the immune homeostasis [70].

*LncRNA-Mirt2.* Mirt2 has different roles in TLR4 signaling mediation, such as MAPD, NF-κB signaling, and MyD88 dependent cascades by inhibiting the oligomerization and ubiquitination of TRAF6; conversely, no effects were observed for TRAF3, TRIF-dependent ubiquitination, and the expression of IFN type I. LPS stimulation activates TLR4 signaling to trigger proinflammatory signaling pathways dependent on the E3 ubiquitin ligase TRAF6. Mirt2 prevents the aberrant inflammation activation, like a potential regulator of macrophage polarization. A recent work demonstrated that *Mirt2* adenovirus-mediated gene transfer could protect mice from endotoxemia-induced fatality and multiorgan dysfunction. These findings identify lncRNA Mirt2 as a negative feedback regulator of disproportionate inflammation [71].

*LincRNATnfaip3*. Located in mouse chromosome 10, in proximity to Tumor necrosis factor α-induced protein 3, in humans *LOC100130476* [91,92] is an early activated NF-kB gene in murine macrophages. In RAW264.7 cell line, it was found upregulated after 30 min and maximum levels were observed at 2–4 h following LPS stimulation (TLR4 activation), while NF-kB inhibitor attenuated its overexpression. Knocking down this LincRNA heavily altered protein-coding genes in RAW264.7, but the majority of the genes affected by lincRNA-tnfaip3 were immune or inflammatory genes like *IL-6*, *TNFaip-3,* and *ICAM-1*. Cross-linking RIP analysis elucidated the physical association with NF-kB subunits; in particular, the lincRNA functions as a bridge binding one of the RNA-binding proteins partners to NF-kB: HMGB-1, under LPS stimulation. NF-kB/HMGB-1 complex recruitment was also analyzed following LPS stimulation and Chip analysis demonstrated a decreased recruitment of this complex to the promoter region of inflammatory gene loci like *TNFaip3* and *IL-6*, when the *LincRNA-TNFaip3* levels were knocked down. On the other hand, overexpression of this lincRNA increased histone H3 modification and transcriptional activation of these genes. In vivo, high lincRNA-Tnfaip3 levels were being experienced in PMPMs, lungs, and spleen of LPS-septic mice models; several inflammatory genes, including *CCL-5*, *Saa3*, *IFNbeta1,* and *CXC-L2* were found upregulated in these districts as well showing an interesting positive correlation [72].

*MALAT1.* Zhao and collaborators, in their reports showed MALAT1 significantly upregulated in LPS-activated THP1 macrophages. MALAT1 was also upregulated in murine RAW264.7 macrophages and in dendritic stimulated cells after low LPS amount stimulation. They found increased expression of inflammatory cytokines TNF-α, and IL-6, but no effects on IL-1β induced by the knockdown of MALAT1. Moreover, MALAT1 regulated NF-kB expression to create a nuclear RNA-protein complex, thus reducing the binding of NF-kB to the promoter targets. *MALAT-1* inhibits NF-kB activity and its target genes, showing a negative correlation between MALAT1 and NF-kB. In particular, MALAT1 in THP1 skin LPS stimulated cells, induced a downregulation of heterodimer p50/p65 activity. Consequently, this mechanism attenuates the expression of TNF-α and IL-6 [73]. In another report, MALAT1 expression is upregulated in LPS and downregulated in IL-4 stimulated macrophages. Human peripheral blood mononuclear cell-derived (PBMC-derived) and human monocytic line THP-1-derived macrophages were used for the experiments. MALAT1 was a direct transcriptional target of LPS-induced NF-κB activation while MALAT1 was inhibited by IL-4 in macrophage models. Therefore, MALAT1 expression was different after LPS and IL-4 stimulations of macrophages. The lncRNA knockdown reduced C-type lectin domain family 16, member A (Clec16a) expression. Clec16a was required for the proinflammatory activation of macrophages [74].

*MIR3142HG*. NGS technology has been used to elucidate lncRNAs function in human lung derived fibroblasts inflammatory response following IL-1β stimulation. In MIR3142HG knockdown in vitro models a significant IL-8 and CCL2 decrease was observed following exposure to IL-1β, showing a positive correlation with the production of these inflammation mediators. This did not occur in IPS fibroblasts. MIR3142HG is *mir3142* host gene, but also contains *mir 146a*, a known positive inflammation and immune response regulator [76,77]. In this cell line, MIR3142HG*/mir146a* was highly downregulated justifying lower cytokine production. Moreover, it was demonstrated that MIR3142HG enhances inflammatory response via NF-kB through the inhibition of NF-kB activator with TPCA-1 [75].

*MORRBID.* MORRBID is known for the numerous roles played in controlling the survival of neutrophils, eosinophils and classic monocytes in response to prosurvival cytokines, in controlling the proapoptotic gene Bim, providing rapid control of apoptosis in response to extracellular survival signals. MORRBID is present in humans and dysregulated in subjects with hypereosinophilic syndrome (HES): a group of blood disorders characterized by high numbers of circulating eosinophils. On these bases, MORRBID could represent a potential therapeutic target for inflammatory disorders [78].

*NEAT1.* Imamura et al. have demonstrated that the *lncRNA NEAT1* (Nuclear Enriched Abundant Transcript 1) is activated by TLR-3. It is an essential lncRNA for the formation of nuclear body paraspeckles, by Toll-like receptor3-p38 pathway-triggered poly I. *NEAT1* is induced by influenza virus and herpes simplex virus infection and facilitates the expression of cytokines, like IL-8. NEAT1 binds to a representative of IL-8 transcription known as SFPQ (proline/glutamine-rich junction factor) and transfers it to paraspeckle bodies, resulting in IL-8 transcriptional activation [79].

*NEAT1v2/eRNA07573.* To identify all the genes modulated in the bacterial infection response, whole-transcriptome analysis was carried out by RNA-seq on HeLa cells 18 h after infection with *Salmonella*. These results, taken together with other RNA-seq analyses on *Salmonella*-infected cells [93,94], enlightened a strict correlation between eRNA upregulation and their associated mRNAs involved in the inflammation pathways. Some ncRNAs were found upregulated. Neat1v2, for example, is a known lncRNA which generates paraspeckles in the nucleus with transcriptional regulators such as NONO and SFPQ [95] and also participates in virus response transcriptional regulation of immunity-related genes [79].

This ncRNA was regulated, due to enhanced stabilization, after the *Salmonella* infection was set and stable. To check the role of Neat1v2, a CRISPR knock out cell line was generated and again infected with *Salmonella*, then compared to both wild type and Neat1v2 KO cells showing an abundance of immune response genes in wild type cells including *TNFsf9*, *CSF1*, *CCL-2*, *NOTCH1,* and *KCTD11*. The same experimental pattern was used with *eRNA07573*, one of the most stabilized enhancer RNAs, comparing KO cells to wild type. RT-qPCR results showed *SLC2A3* and *SLC2A14* genes expression to be highly downregulated in *eRNA0757* KO cells. These genes are positioned adjacently to *eRNA07573* locus and code for glucose transporters, required for *Salmonella* growth in host cells [80].

*NKILA*. An important regulator of the proinflammatory response is lncRNA NKILA (NF-kB-interacting lncRNA), a critical repressor to protect the endothelium from inflammation. NKILA is able to positively mediate the expression of *KLF4*, an anti-inflammatory atheroprotective regulator in endothelial cells (ECs), by a NF-kB-mediated DNA methylation mechanism [81].

*NRIR*. Toll-like receptor 4 belongs to recognition receptor family (PRR) which main function is the activation of innate immune response. It acts in two ways: via MyD88, that leads to the cytokine release, or via TRIF receptor activation that, above all, increases INF-beta through IFN I pathway. Investigating these interactions after LPS activation of TLR4, RNA seq has been carried out to assess if lncRNA profile was altered in primary human monocytes. Several lncRNA resulted upregulated or downregulated at different time points after stimulation but then the results were correlated to type I IFN response and lncRNA NRIR appeared to be involved [82].

*PACER.* lncRNA (PACER) is involved in preventing a repressor complex (NF-κB p50) to associate with promoter (p300 HAT). PACER interacts with p50, which can form both active heterodimers with p65/RelA, and inactive p50/p50 homodimers that lack transcription activation domains present in p65/RelA. PACER regulates COX-2 expression by not physically interacting with this subunit. Human mammary epithelial cells and human histiocytic lymphoma monocyte cell line U937 were used for this study. The main function of PACER lncRNA is to facilitate the assembly of the RNA polymerase II preinitiation complexes, thus representing a new potential transcriptional control for COX-2 in inflammation and cancer [53].

*PTPRJ-as1.* PTPRJ/CD148 is a tyrosine phosphatase protein and acts as tumor suppressor. PTPRJ-as1 was highly expressed in macrophages after LPS stimulation and other Toll-like receptor ligands, while poorly expressed after CSF-1 treatment. CD148 was also expressed in macrophages. PTPRJ-as1 originates from *PTPRJ* locus in macrophages and is transiently induced by Toll-like receptor ligands, with a course similar to *PTPRJ*. The putative transcription factor binding sites are identified in the promoter region of *PTPRJ*. Moreover, *PTPRJ* expression is upregulated in epithelial cells where it plays a negative role in the regulation of cell proliferation; in contrast *PTPRJ* is a positive activator of B cells and monocytes by dephosphorylating the negative regulatory carboxyterminal tyrosine of Src family kinases (SFK) [83]. The precise mechanism involved in the regulation of inflammation by *PTPRJ* remains to be elucidated.

*THRIL*. Innate immune cells such as macrophages and neutrophils are present in inflamed joints due to TLR signaling pathway. LncRNA THRIL (TNF and immunoregulatory lincRNA) related to nuclear RNPL (hnRNPL), was recently identified for the induction of TNFα acting through a ribonucleoprotein (RNP) complex with hnRNPL (protein involved in the stress responses). The THRIL-hnRNPL complex has been shown to bind the TNF promoter and induce its expression after TLR-2 activation. THRIL is a key player in regulating TNFα expression in macrophage cells [55].A brief summary of the most studied lncRNAs and their DNA modulation role in immunity is schematized in Figure 2.

## 6. Expression and Function of LncRNA in Adaptive Immunity

Lymphocytes (T and B cells), primary cellular mediators of the adaptive immune system, are known to express many lncRNAs and play crucial roles in their development, specific lineage differentiation, and activation. Below we report the new emerging evidence linking individual lncRNAs with T and B cells function and differentiation (Table 2).

*GAS*5. Growth arrest-specific transcript 5 (GAS5) is a 5′TOP RNA (5′ terminal oligopyrimidine sequence). Its translation is correlated with mTOR pathway since in mammalian lymphocytes mTOR inhibition leads to a severe translation impairment in mRNAs encoding ribosomal and translation-related proteins [104]. mTOR pathway inhibition with rapamycin and consequent cell growth arrest increased GAS5 levels [105] that are also required for normal T-cell growth [96]. It has been demonstrated that GAS5 downregulation has a protective effect on primary leucocytes from growth inhibition induced by mTOR antagonists in human PBMNCs. These data, also confirmed in Leukemic T cells, show that this ncRNA is required for the proper action of mTOR inhibitors [96]. Moreover, GAS5 has been associated with the repression of growth-arrest induced by starvation and glucocorticoid receptors. It binds to DBD region and prevents the DNA binding and consequent receptor-activated effects. Since glucocorticoids are frequently used in immune-related diseases due to their potent immunosuppressant effect, high GAS5 expression might play a role in enhancing autoimmunity disorder by inhibiting their action [97].

*lincR-Ccr2-5′AS*. After RNA-seq analysis on 42 different T cell subsets, several lincRNA-encoding genes were detected during differentiation and development of T cells. Most of them were preferentially expressed in TH1 or TH2 cells with a consistent onset at 48–72 h and expression plateau at 1–2 weeks. One of them was lincR-Ccr2-5′AS that is associated to genes involved in chemokine-mediated signaling pathway, especially in Th2 cells. To better understand its activity, it was knocked down in those cells. No change in IL-4 production was recorded compared to controls. On the other hand, genes in close proximity to this lincRNA like *Ccr1*, *Ccr2*, *Ccr3,* and *Ccr5* were downregulated. CD45^+^ cells with low lincRNA expression had their migration capability to the lungs of C57BL/6 mice significantly impaired. Moreover, it was also correlated with GATA-3, and together they are regulators of T cell immunological function and differentiation [50].

*lncRNA Rmrp.* This lncRNA belongs to mitochondria RNA Processing Endoribonuclease Complex (MRP). Mutated *RMPR* is associated in humans to cartilage-hair hypoplasia, defective immunity, predisposition to lymphoma, and neuronal dysplasia of the intestine [106,107]. TH17 differentiation is under the control of RORγ, a nuclear receptor that regulates cytokine based transcriptional network. Genes involved are TH17 cytokines like *IL-17A*, *IL-17F*, *IL-22*, *IL-23R*, *IL-1R1* but also *CCR6* [108]. Liquid chromatography-tandem mass spectroscopy evidenced the high interaction between DDX5 RNA helicase and ROR γ-t in TH17 cells. In vitro and in vivo results showed DDX5 as a selective regulator in the Th17 effector program in steady-state and during inflammation. In particular, it was associated with enhanced cytokine production. Then putative lncRNAs involved in DDX5-ROR γ-T axis was investigated through RIP-qPCR. *LncRmrp* was the most enriched lncRNA associated with DDX5 complex. Depletion of *lncRmrp* reduced *IL-17a* and *IL-17F* mRNAs both in vitro and in vivo, similarly to that obtained after DDX5 depletion. Furthermore, the lncRNA action required the presence of DDX5 [98].

*NeST.* NeST is located adjacent to human and mouse INF-γ encoding gene. Its most abundant 914 nucleotides splice variant is expressed in CD4+, CD8+ and in NK cells [100,101]. This lncRNA has the capability to reproduce the phenotype associated with *Tmevp3* locus. *Tmevp3* is involved in Theiler’s virus clearance. Its phenotype is related to the lack of Theiler’s virus infection resolution but also immunity to *Salmonella enterica* lethal infection, IF-γ inducible synthesis in CD8+ T cells in B10.S mouse models. It has been demonstrated that transgenic expression of NeST RNA reduced *Salmonella* pathogenesis, preventing Theiler’s virus clearance and increasing the INF-γ expression and synthesis in CD8+ T cells in the same animal models. NeST is mainly located in the nucleus and acts at the transcriptional level by interacting with chromatine, directly increasing H3K4me3 activation in CD8^+^ T cells and inducing IFN-gamma gene after WDR5 binding. NeST, a long noncoding RNA, controls microbial susceptibility and epigenetic activation of the IFN-gamma locus [99].

*NRON*. NFAT, nuclear factor of activated T cells, is phosphorylated in resting cells. This phosphorylation [109], by CK1 and GSK3 and DYRK results in the cytoplasmic localization and inactivation of this nuclear factor due to masking of the nuclear localization sequence and the exposure of nuclear export sequence. T-cell stimulation reverts this phosphorylation and causes NFAT nuclear accumulation and target gene transcription [11,110]. It has been shown that phosphorylated NFAT1 is contained in RNA-protein complex together with its three kinases, lincRNA NRON and IQGAP1 scaffold protein in Jurkat T cells. Moreover, IQGAP1 and NRON had synergic activity in NFAT regulation since their knockdown led to a significant increase in NFAT dephosphorylation and nuclear translocation, higher if compared to single knockdowns alone. After the lincNRON knockdown in Jurkat cells, NFAT-dependent genes were analyzed and IL-2 production was remarkably higher in this model after PMA and ionomycin stimulation compared to the controls. Overall, this scaffold contributes to the NFAT inactivation and stabilization in the cytoplasm [102,111].

*NTT.* Noncoding Transcript in T cells is a particular lncRNA expressed in CD4+ T cells and PBMCs nucleus after HIV peptides stimulation [112]. It is located at chromosome 6q23-q24 in close relationship to *IFNGR1*, *TNFAIP2*, *PBOV1,* and *MYB* genes involved in immunity, hematopoiesis and proliferation [113]. The role of this lncRNA in T cells remains unclear, but it has been shown that in human monocytic cells, macrophages and THP-1 it is regulated by c/EBPβ. Furthermore, after its knockdown in THP-1 cells it has high impact on *PBOV1* expression. RIP and ChIP confirmed this interaction to happen through hnRNP-U protein interacting with *PBOV1* promoter after NTT binding. *PBOV1* overexpression in THP-1 caused G1 phase stop of the cell cycle and was associated to macrophage differentiation. In freshly diagnosed Rheumatoid Arthritis patients, PBMCs, C/EBPβ, NTT and *PBOV1* expression was found highly upregulated compared to healthy controls. The levels were decreasing with time and treatment. In the same patients, *IFNGR1* and *TNFAIP3* levels were downregulated. These data combined with patient’s follow-up data analysis suggested C/EBPβ/NTT/PBOV1 axis higher expression level to be associated with inflammatory status and to require more pharmacological attention [103].

## 7. Expression and Function of LncRNA in Diseases Related to Inflammation and Immunity

It is currently known that lncRNA dysregulation can exert an impact on the pathogenesis of autoimmune diseases including, rheumatoid arthritis (RA), psoriasis, Sjögren’s syndrome (SS), Crohn’s disease (CD) (Table 3).

*ANRIL*. ANRIL has single nucleotide polymorphisms (SNPs) in its chromosomal region, and these mutations are associated with susceptibility to cardiovascular disorders [114]. SNPs are associated to cardiovascular disease [129]: evidencing minor mutations and genetic variations such as single nucleotide polymorphisms (SNPs) can have a small effect on the function of lncRNAs. Another line of evidence reported the association of ANRIL gene with coronary artery disease, intracranial aneurysm, and type 2 diabetes. In aortic soy muscle cells, artery disease models, CNS3 preserved sequence in ANRIL was enhanced after reporter genetic experiment analysis.

*BANCR.* This lncRNA has been associated with poor prognosis in melanoma cells and connected to MAPK pathway [115]. It was found upregulated in melanoma biopsy tissues showing a direct correlation with tumor stage, associated with bad prognosis in malignant melanoma patients. In five different melanoma cell lines, BANCR was found remarkably upregulated compared to controls. Knocking down this lncRNA resulted in proliferation impairment in both cell lines and in vivo xenograft on nude mice. ERK1/2, Raf-1 and JNK resulted inactivated after BANCR knockdown (no effect showed on p38 MAPK levels), and its overexpression reverted the phenotype, increasing their expression even if ERK1/2 and JNK inhibitor were used. In lung cancer, BANCR was found downregulated both in tissues and cell lines compared to adjacent noncancerous tissue and human bronchial epithelial cell line. Functional assays revealed that H446 and SPC-A1 cells proliferation and migration was increased if BANCR was downregulated, the same happened in vivo. BANCR overexpression caused p38 MAPK and JNK inactivation in those cell lines but did not affect ERK1. Silencing BANCR, on the other hand, rescued the phenotype [116].

*Fas-AS1*. Fas, member of TNF receptor family, and its ligand exert a major role in apoptosis extrinsic pathway [130]. Defects on its actions are associated with autoimmune diseases and lymphoma [131,132,133]. It is known that cancerous cells evade apoptosis also by exploiting *Fas* mRNA alternative splicing, generating soluble decoy receptors (sFAS) not transducing the signal after FasL binding. This is correlated with the severity of tumor and poor clinical outcome, especially in non-Hodgkin lymphoma [134]. FAS-AS1 lncRNA is an antisense RNA transcribed from *Fas* gene and was identified to have a role in the sFas regulation. In lymphoma cell lines FAS-AS1 expression was lower compared to the healthy B cells, but, on the other hand, sFas levels were higher. Fas-AS1 expression is repressed by EZH2-mediated H3k27 tri-methylation since inhibiting H3K27 was increasing its levels. Furthermore, FAS-AS1 lncRNA binds and interferes with RNA binding protein RBM5 a known enhancer of *Fas* mRNA alternative splicing [117].

*HOXD-AS1*. Bioinformatic studies started with Affimetrix array reanalysis and improved with a new generated pipeline about lncRNA correlation in neuroblastoma aggressiveness and relapse, enlightened a 159 lncRNA signature. HOXD-AS1 was taken as the most promising candidate related to oncogenesis since it is located in the *HOXD* cluster and *HOX* genes are correlated with malignancy. PI3k and MAPK signaling pathways were investigated since they mediate the differentiation and survival of SH-SY5Y cell line after retinoic acid stimulation. Retinoic acid is commonly used in NB therapy since it can stop the growth and induce differentiation. Retinoic acid caused HOXD-AS1 induction likely via the PI3k pathway. Moreover, this lncRNA was correlated to inflammation, that strengthened tumor growth with favorable crosstalk with tumor microenvironment. In particular, HOXD-AS1 target genes encode for cytokines such as CX3CL1, CCL20, TNF, GDF15 [118].

*HOTAIR.* HOTAIR was the first lncRNA to be characterized and has been demonstrated to interact with the polyphonic repressive complex 2, which regulates the state of chromatin and histone demethylase (lysine 1 specific demethylase). There was a strained activation of Wnt/β-catenin pathway by HOTAIR declining. Mao et al. suggested that silencing lncRNA HOTAIR and inhibiting Wnt/β-catenin pathway reduced synovial inflammation and synoviocytes proliferation and promoted apoptosis in osteoarthritis rats [119].

*Lnc13*. Lnc13 genomic region is highly conserved between human and mice. Its expression was confirmed at low levels in human U937 myeloid cell line and appeared to be present in human intestinal paraffin-embedded sections. This lncRNA seemed quite tissue specific, preferentially expressed in intestinal villi and lamina propria, and its correlation with celiac disease (CeD) was then investigated. It appeared to be remarkably downregulated in CeD patient biopsies compared to healthy controls. Its subcellular localization was nearly completely nuclear in macrophages and mononuclear cells of the lamina propria. After mouse macrophages (DMDMs/iBMMs) and human U937 stimulation with LPS, *lncRNA13* levels were significantly decreased compared to untreated controls. The downregulation was related to Myd88 and NF-kB pathway activation, constitutively active in celiac disease patients [120]. After silencing this pathway Lnc13 levels were upregulated. Moreover, its activity was associated to a wide panel of genes involved in inflammation. It inversely correlates with Traf2, Stat1, Stat3, Tnfsf10, Il2ra, Ccl12, Myd88, Csf3, and Il1ra after LPS stimulation. In celiac disease biopsies TRAF2, STAT1, IL1RA, and MYD88 were found upregulated, and after Lnc13 upregulation, their expression was reduced, while Lnc13 knockdown was reverting the phenotype. Further analysis on this mechanism showed its interaction with chromatin and the multifunctional protein hnRNPB. Lack of Lnc13 impaired the maintenance of intestinal mucosal immune homeostasis and contributed to celiac disease [121].

*Linc-Maf-4.* Multiple sclerosis is a CNS disease that causes axonal damage and demyelination by autoreactive lymphocytes prompting an inflammatory response and leading to neurologic disfunction [135,136,137,138].

Several studies demonstrated CD4+ T cells, TH1, and TH17 involvement in the disease course [139,140,141,142]. PBMCs from 34 patients with acute MS were analyzed with RNA microarrays and linc-MAF-4 was found significantly upregulated while MAF was downregulated. Then Linc-Maf-4 and *MAF* mRNA levels were evaluated on TH1/2/17 and T reg cells generated from CD4+ naive T cells from MS patients. During TH1 stimulation Linc-Maf-4 levels were higher, while following TH2 stimulation *MAF* mRNA levels were increased; this inverse correlation was also observed after knocking in and knocking down this lincRNA in naive CD4+ cells. Moreover, those cells, after lincRNA knockdown, showed a significant percentage decrease of IFN-γ producing T cells and increase IL-4 producing cells suggesting that linc-MAF-4 is required for TH1 differentiation. Surface receptor studies enlightened that this lncRNA also promotes CD4+ T cells activation. In vivo study on MS patients also correlated higher level of lncRNA to an increased number of lesion and relapses. Overall linc-MAF-4 seems to be associated with the pathogenesis and the severity of multiple sclerosis [122].

*LncRNA-CMPK2*. Annotated as AC017076.5, this lncRNA was named -CMPK2 for its position downstream of the protein-coding ISG *CMPK2* and is greatly induced after INF-α stimulation in Huh7.5 cells. An INF-stimulated response element is located upstream this lncRNA. Its shorter isoform is the most abundant and it is located mainly into the nucleus as an independent non-protein-coding transcript. To go further inside the correlation between INF-α and this lncRNA, Jak-STAT pathway was investigated. First, roxolitinib, a JAK inhibitor, was used and reverted nearly completely the lncRNA overexpression following the stimulation. This was also happening when inhibiting STAT2. These data demonstrated that the lncRNA is induced by Jak/STAT pathway activation following INF stimulation, and this did not occur after TNF-α/NF-kB pathway activation. These results were confirmed in a wide panel of human cell lines such as HEK293T, H1975, HeLa, Jurkat, THP1, and primary keratinocytes. The role of this transcript in INF response pathways was further examined knocking it down throughout shRNA constructs and analyzing the outcome of HCV infection during INF-α treatment. Viral genomic RNA resulted significantly reduced in the lncRNA knockdown cell line compared to wild type ones. Moreover, in the lncRNA knockdown, ISGs levels, such as *ISF15*, *CXCL10*, *IFIT3*, *IFITM1* were upregulated suggesting that this lncRNA might act through regulation of nuclear events. In vivo, lncRNA-CMPK2 was found upregulated in liver samples from HCV-infected patients showing once more correlation with the viral infection [123].

*lncRNADQ786243*. DQ786243 function has been analyzed in patients with Crohn’s disease (CD) and in healthy controls. The expression of DQ786243 was closely correlated with the expression of the binding protein of the cAMP response element (CREB), important for TCR response element activity in the forkhead box P3 (Foxp3): principal transcription factor involved in the regulatory T cells (Treg) development. It was hypothesized that Foxp3 could be less expressed in CD patients compared to controls. Moreover, it was confirmed that *Foxp3* mRNA levels were comparable in CD patients and in controls. No close interactions have been found between the expression of *Foxp3* and *CREB*. DQ786243 was strongly correlated with *Foxp3* and *CREB*. All the three mRNAs were correlated with CRP, an important serum marker of inflammation. This suggested that DQ786243 could contribute to CD development, but the malfunction of DQ786243 in the CD patient could not be ruled out. On the Jurkat cell line, DQ786243 upregulated by transfection increased the phosphorylation ratio of *CREB*. *CREB* itself did not appear to be the mediator of DQ786243 responsible for Foxp3 upregulation. Hence, DQ786243 could positively influence Foxp3’s expression [16].

*LncRNA-p21.* LncRNA-p21 inhibits NF-kB signaling by sequestering RelA (the p65 subunit of NF-kB) in T cells in patients with Rheumatoid arthritis (RA). The expression of LincRNA-p21 was shown to be upregulated by methotrexate via a DNA-dependent protein kinase catalytic subunit–dependent mechanism [124].

*lncRNA RMPR.* Mutations in the transcribed region or in the promoter of RMRP1 genes in humans lead to a rare autosomal recessive disorder, cartilage-hair hypoplasia. It is characterized by skeletal dysplasia, hypoplastic hair, neuronal dysplasia of the intestine, skeletal dysplasia, and defective immunity with and early childhood onset. This is frequently associated with autoimmune diseases in joints and kidneys and hematological abnormalities [98,106,107,125].

*lncRNA SAS-ZFAT*. *ZFAT* is a zinc finger related gene in the auto-immune thyroid disease (AITD). *ZFAT* was found as one of the susceptibility genes in 8q23-q24 through a sample association analysis on a total of 515 AITD patients and 526 controls. In particular, it was found that the T allele single nucleotide polymorphism (SNP), Ex9b-SNP10, is an SNP found in the *ZFAT* intron, the 3′-UTR of the truncated form of *ZFAT (TR-ZFAT)*. The study carried out in peripheral blood lymphocytes reported that *SAS-ZFAT* is expressed exclusively in CD19^+^ B cells and the expression levels of *SAS-ZFAT* and *TR-ZFAT* seem to be correlated with the *ZFAT*-allele associated with Ex9b-SNP10, respectively, inversely and positively. Hence, Ex9b-SNP10 is critically involved in the regulation of SAS-ZFAT expression and this expression causes a reduction in TR-*ZFAT* expression increasing susceptibility to AITD [126].

*MALAT1.* Recent studies suggest the involvement of MALAT1 (transcript 1 of lung adenocarcinoma associated with metastasis) in p38 MAP kinase pathway [143]. In particular, a MALAT1-MAPK p38 crosstalk is described in diabetes mellitus phatogenesis. MALAT1 might be able to activate p38 MAPK, since after its downregulation with siRNA, p38 phosphorylation reduced, as well. MALAT1 could also increase the expression of IL-6 and the generation of reactive oxygen species [144]. Moreover, MALAT1 was involved in inflammation and epigenetic regulation of diabetic retinopathy (DR), where the global inhibition of DNA methyltransferases induced significant increases of the levels and associated inflammatory transcripts in HRECs (human retinal endothelial cells) [127].

*MORRBID*. MORRBID plays several roles; it is involved in the regulation of *Bim* pro-apoptotic gene transcription, controls the life of neutrophils, eosinophils and monocytes in response to prosurvival cytokines. The role of MORRBID was investigated in MORRBID-decifient mice models, highly sensitive to bacterial infection (*L. monocytogenes*) and protected from eosinophilic-guided allergic lung inflammation. Results showed MORRBID to be a critical regulator of eosinophils, neutrophils, and Ly6Chi monocytes, because their levels were drastically reduced in blood stream. Moreover, MORRBID acts intrinsically within the cell because deficient BM cells have a significant defect in the generation of short-lived myeloid cells. The expression of the MORRBID pattern, *Bim*, was shown to be very high in cell lines in MORRBID deficient mice and was not dysregulated in other populations of myeloid and lymphoid cells. Hence, MORRBID regulates the transcription of its neighbor proapoptotic gene, Bim, promoting the enrichment of the PRC2 complex in the *Bcl2l11* promoter to maintain this gene in a balanced state. Therefore, MORRBID, through the expression of the *Bcl211* gene, can regulate the lifespan of myeloid cells. IL-3, IL-5, and GM-CSF cytokines are known to promote the survival of eosinophils, neutrophils, and Ly6. BM-derived eosinophils lysates showed a possible loss of MORRBID expression and an increase in Bc2l11 levels. On the contrary, the addition of IL-5, IL-3, or GM-CSF induced the expression of MORRBID, accompanied by the repression of *Bcl2l11*. Similarly, stimulation of the ex vivo β-chain cytokines, but not the stimulation of G-CSF, significantly induced MORRBID and a corresponding repression of *Bcl2l11* in neutrophils and Ly6Chi monocytes. Importantly, MORRBID deficient neutrophils were unable to inhibit the expression of Bcl2l11 after the addition of β-chain cytokines. These results suggested that β-chain cytokines repress the expression of Bcl2l11 in short-lived myeloid cells in a MORRBID-dependent manner. Therefore, in these highly inflammatory cells, changes in MORRBID levels provide a locus-specific regulatory mechanism that allows rapid control of apoptosis in response to extracellular survival signals. MORRBID is present in humans and dysregulated in patients with hypereosinophilic syndrome representing a potential therapeutic target for inflammatory disorders characterized by short-lived aberrant myeloid cell duration [78].

*NRIR.* Its levels were examined in systemic sclerosis patients, compared to healthy controls, showing consistent upregulation, that correlated with increased INF pathway activation [82].

*OLA1P2.* This particular lncRNA function has been investigated on colorectal cancer (CRC) treated with aspirin. Aspirin has a protective action against several cancers, especially CRC [145,146,147,148] due to its inhibiting COX-1-dependent platelet function [149], PP2A enzyme activity and p-Beta-catenin downregulation [150], ERK-mediated signaling suppression [151], NF-κB pathway inhibition [152], and mTOR signaling reduction [153]. Under these circumstances, microarray analysis on primary CRC cells showed more than 20 fold upregulation of *OLA1P2* compared to untreated cells. These results were also confirmed in several other cancer types, such as oral, gastric, and colon cancer. To identify OLA1P2 expression responsible factors, all the nuclear proteins interacting with a biotin-labeled OLA1p2 promoter were pulled down and analyzed through mass spectrometry. In particular, *USF1*, *PAX6*, *SMAD3*, *FOXD3*, *TP53*, *NCOA1*, *NCOA3*, *MEF-2* levels were greatly increased in the nucleus after aspirin treatment and further immunoblot validation. Moreover, after selective silencing of all these genes, *FOXD3* was the one who showed direct correlation and strong binding activity with the lncRNA. In OLA1p2 knockdown primary cells, a significant enrichment of the STAT3 signaling pathway was observed. Physical interaction between the lncRNA and phosphorylated STAT3 (Tyr705) was then confirmed by RIP and FISH analysis. It was also shown that this interaction prevented phosphorylated STAT3 homodimer formation. OLA1P2 antiproliferative and antimetastatic effects were confirmed in vitro. OLAP1P2 silencing in immunodeficient mice greatly reverted the antimetastatic aspirin effect. As expected, in CRC clinical cancer tissues in treatment with aspirin, it was reported a significant negative correlation between FOXD3 and phosphorylated STAT3 (tyr705) proteins, while low levels of *OLA1p2* were correlated with the progression of the tumor and to poor prognosis [128].

## 8. Conclusion and Future Perspectives

LncRNAs have recently emerged as potential immune system regulators and responsible for the development of autoimmune diseases. Widespread attention on lncRNAs is a rather recent phenomenon, nevertheless some evidence confirms the role of lncRNA in autoimmune diseases. The inflammatory response is an extremely complex biological function, and lncRNAs can be activated or regulated inappropriately leading to the disease onset. However, highlighting the functional roles for lncRNA in the context of immune disease remain challenging due to lack of protein products, tissue-specific expression, low complexity expression levels in splice shapes, and lack of interspecies conservation. In this review, we briefly sought to clarify the role of lncRNAs in regulating gene transcription during the inflammatory response, both innate and adaptive, and the main aspects in the immunity related diseases development. LncRNAs act through different mechanisms that include the lead of chromatin-modifying complexes to specific genomic loci, providing molecular scaffolds and modulating transcriptional programs. LncRNAs dysregulation is known to be associated with a number of human diseases and they can improve or suppress the inflammatory response specifically for a gene and for a specific time. Although, it is clear that lncRNAs are expressed and regulated during immunity, by targeting various metabolic pathways in different manners. Furthermore, lncRNAs are known to be poorly conserved evolutionarily, however some of them such as MALAT1, IL7-AS, FIRRE, are expressed in both human and murine species. Therefore, in depth knowledge of their domains could be used to test their functional relevance in the development of inflammatory disease.

As we mentioned, the family of TLR (Toll-like receptor) molecules is composed by pattern recognition receptors important for immune responses and are implicated in very different inflammatory diseases. Although the upstream signaling components mediating the TLRs pathogens detection are clear, we know very little about the downstream transcription cascades that directly control the expression of specific genes. Research is going toward new single-cell genomic technologies to identify regulatory regions and large noncoding RNAs (lncRNAs) contributing to the regulation of the immune system after pathogens exposure. LncRNAs have long been considered as transcriptional noise but growing evidence points to their involvement in various physiological processes. From these assumptions, they can potentially be identified as new specific biomarkers, acting as diagnostic and prognostic indicators of the disease. The lncRNAs numerically constitute the majority of noncoding RNAs within the mammalian genome. Understanding that noncoding RNA is hiding inside the noncoding part of the DNA has brought breath, hope, and knowledge. The finer understanding of these mechanisms will accompany us in the coming years in our efforts for basic research and clinical management of patients. These molecules, in fact, could serve as biomarkers for the early diagnosis of inflammatory diseases and for the evaluation of the predisposition both for the development of the pathology and for the predisposition to complications. Moreover, they can be exploited in precision medicine and specific targeted therapy, since restoring the altered lncRNAs physiological levels might represent a powerful tool in the future to avoid or delay inflammatory disease onset or even prevent their worsening, cooperating with standard therapy. In conclusion, lncRNAs could represent a potential future application for novel therapeutic strategies against autoimmune diseases, chronic inflammations and infections. Future studies will be needed to deepen our understanding about immune functions of lncRNAs and translational applications.

## Figures and Tables

**Figure 1 ijms-22-01741-f001:**
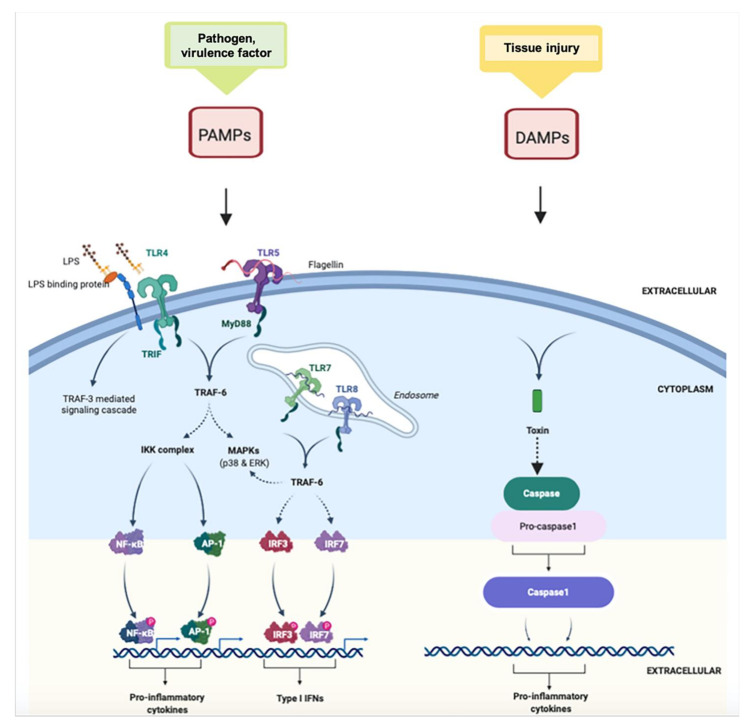
Regulation of inflammatory response. Dotted lines indicate presumable mechanisms, normal lines indicate known activation. PAMPs: molecular patterns associated with pathogens, DAMPs: molecular patterns associated with danger, LPS: Lipopolysaccharide, TRIF: TIR-domain-containing adapter-inducing interferon-β, TRAF: TNF receptor-associated factor, Myd88: Myeloid differentiation primary response 88, NF-kB: nuclear factor kappa-light-chain-enhancer of activated B cells, IKK: IκB kinase, MAPK: mitogen-activated protein kinase, AP: Activator protein, IRF: Interferon regulatory factors.

**Figure 2 ijms-22-01741-f002:**
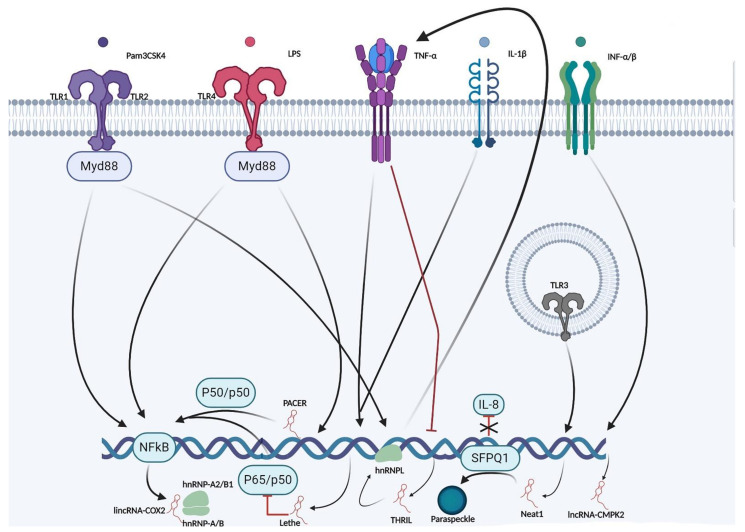
Examples of lncRNA roles correlated to immunity: a schematic summary about the most studied lncRNAs involved in DNA modulation discussed above. Normal arrows represent an interaction while red ones an inhibition. IL: Interlukin, hnRNP: Heterogeneous nuclear ribonucleoproteins.

**Table 1 ijms-22-01741-t001:** LncRNAs roles and mechanisms in innate immunity.

LncRNA	Model	Function	Main Results	References
***ANRIL***	Human umbilical vein endothelial cells (HUVECs)	Regulate cell proliferation	Regulated by TNF-α and NF-κB	[58]
***AS-IL1α***	Mouse bone marrow derived macrophage	Regulates IL-1α transcription	Facilitates the recruitment of RNAP-II (RNA Polymerase II) to the *IL-1α* locus	[59]
***FIRRE***	Human macrophages (U937) Human intestinal epithelial cells (SW480) Mouse macrophages (RAW264.7) Jurkat cell	Regulates expression of several inflammatory genes	Demonstrates NFkB-dependent expression; interacts with hnRNPU to regulate mRNA; increasing the *VCAM1*	[60]
***H19***	Mouse hematopoietic stem cells (HSC)	Maintains HSC quiescence	Regulates the Igf2-Igfr1 (insuline-like growth factor) pathway through the translocation of FOX3 to the cytoplasm	[61]
***IL-1b-RBT46***	Human monocytes (THP-1)	Act in *IL-1* and *CXCL8* regulation	NF-kB dependent expression	[62]
***IL7-AS***	Human monocyte (THP-1) cells, and LPS-stimulated mouse (RAW 264.7)	Regulates *IL-6* expression	NF-kB dependent expression	[63]
***Lethe***	Mouse embryonic fibroblasts (MEF = lines mouse macrophages (RAW 264.7) and bone marrow derived macrophages)	Regulates NF-kB mediated inflammatory genes, ROS production, and *NOX2* gene expression	Interacts with the RelA (p65) subunit of NF-kB	[64,65]
***LincRNACOX2***	LincRNA-Cox2 deficient mice; mouse macrophages (RAW 264.7 and primary peritoneal)	Regulates the expression of NF-kB and several immune genes	Functions as an eRNA to regulate the activity of the *COX2 *(Cyclooxygenase) gene but also demonstrates in trans-regulation of immune-associated genes in vivo. Interacts with the SWI/SNF complex to regulate the assembly of NF-kB subunits and chromatin remodeling	[15,66]
***Lnc-DC***	Human dendritic cells	Regulates DC differentiation	Interact with STAT3 to prevent dephosphorylation of Y705 by SHP1(Src homology region 2 domain-containing phosphatase-1)	[67]
***LincRNA-EPS***	Mouse bone marrow derived macrophages	Inhibits IRGs expression and represses the inflammatory response	Interacts with hnRNPL via a CANACA motif in its 3′ region and regulates nucleosome positioning at IRG promoters	[62]
***LncHSC-1/2***	Mouse hematopoietic stem cells	LncHSC-1 regulates myeloid differentiation and LncHSC-2 cell self-renewal and differentiation	LncHSC-1 is involved in myeloid differentiation LncHSC-2 recruits the hematopoietic TF E2A to its binding sites	[68]
***Lnc-IL7R***	Human monocytes (THP-1)	Regulates the expression of the inflammatory mediators IL-6, IL-8, E-selectin, and VCAM-1(Vascular Cell Adhesion Molecule)	Regulates deposition of *H3K27me3* at the promoters of the E-selectin and *VCAM-1* genes	[54]
***LncITPRIP-1***	Huh7, Huh7.5, Huh7.5.1-MAVS, FL-neo, HEK293T cells	Involved in the activation of the innate immune response	Binds to the C-terminus of MDA5 (melanoma differentiation-associated protein 5) and promotes its oligomerization to enhance IFN signaling and production	[69]
***Lnc-Lsm3b***	Mouse peritonea macrophages RAW, L929 and HEK293T cell lines	Inactivates RIG-1 innate activity and type I IFNs production	Acts as a regulator of innate immune response	[70]
***LncRNA-Mirt2***	Peritoneal macrophages (C57BL/6 mice) HEK293T and RAW264.7 cells	Regulates macrophage polarization and aberrant inflammatory activity	Inhibits the K63-ubiquitination of TRAF6	[71]
***LincRNA-Tnfaip3***	Mouse macrophages (RAW 264.7 and primary mouse peritoneal macrophages)	Regulates the expression of several NF-kB inflammatory genes	Increased histone H3 modification and transcriptional activation	[72]
***MALAT1***	Human monocytes (THP-1), mouse macrophages (RAW 264.7)	Regulates the expression of inflammatory genes Regulates LPS-mediated M1 macrophage activation and IL-4-mediated M2 differentiation	Interacts p50/p65 subunit to inhibit NF-kB DNA binding activity. Demonstrates *Clec16a*-dependent expression and regulation of mitochondrial pyruvate carriers	[73,74]
***MIR3142HG***	Human lung fibroblasts (primary) IPS fibroblasts	Regulates *CCL2* and *IL-8* mRNA and protein release Downregulates *MIR3142HG/mir146a* interaction	Demonstrates NF-kB-dependent expression	[75,76,77]
	Human and mouse monocytes, neutrophils, eosinophils	Controls the survival of inflammatory cells	Regulates the proapoptotic gene *Bim*(Bcl-2-like 11)	[78]
***NEAT1***	Human epithelial cells (A549 cell line) and HeLa cells	Regulates IL-8 expression	Binds as SFPQ (proline/glutamine-rich junction factor) and transfers it to paraspeckle bodies, resulting in *IL-8* transcriptional activation	[79]
***NEAT1v2/eRNA07573***	HeLa cells	Regulates expression of immune-associated genes and response to antibacterial defense	Inhibit levels of the exosome/NEXT components	[80]
***NKILA***	Endothelial cells (ECs)	Mediates the expression of *KLF4*	Mediates the expression of *KLF4* by NF-kB mediated DNA methylation mechanism	[81]
***NRIR***	Human monocytes	Regulates the expression of several interferon-stimulated genes and protein release of CXCL10 and CCL8	Demonstrates type I IFN-dependent expression	[82]
***PACER***	Human mammary epithelial cells and monocyte/macrophage cell lines	Promotes *COX2* expression	Interact with p50 NF-kB subunit of *COX2* promoter to enable p300 HAT in order to facilitate the assembly of the RNA polymerase II	[53]
***PTPRJ-as1***	Mouse macrophages and RAW 264.7	N/A	N/A	[83]
***THRIL***	Human monocytes (THP-1)	Regulates the expression of the innate-associated mediators *TNF-α*, *CCL1*, *IL-8*, *CSF1*, and *CXC10*	Forms a functional lncRNA-hnRNPLcomplex in order to regulate TNF-α	[55]

**Table 2 ijms-22-01741-t002:** LncRNAs role and mechanisms in adaptive immunity.

lncRNA	Model	Function	Main Results	References
*GAS5*	T-cells. Leukemic T-cells	Involved in mTOR (mammalian target of rapamycin ) and glucocorticoid response pathways	Required for mTOR antagonist response in leukemic T cells and represses growth arrest and starvation induced by glucocorticoid receptor	[96,97]
*lincR-Ccr2-5′AS*	Different T-cells subsets especially TH2(T-Helper)	Correlated with chemokine response	Positive correlation with *Ccr1*,*2*,*3* and *5* genes. Increases CD45+ cells migration	[50]
*lncRNA Rmrp*	TH17	Correlated with cytokines response	Direct correlation with IL17a and f expression	[98]
*NeST*	CD8+ T cells	Correlated with INF-γ response	Prevents Theiler’s virus clearance, stimulates INF- γ response	[99,100,101]
*NRON*	Jurkat	Correlated with NFAT-dependent genes	IL-2 production negatively correlated with the lncRNA after stimulation	[102]
*NTT*	Human monocytic cells, macrophages and THP-1	Involved in macrophages differentiation	*C/EBPbeta/NTT/PBOV1* axis increases inflammatory status	[103]

**Table 3 ijms-22-01741-t003:** LncRNAs role in immunity related disease.

LncRNA	Model	Pathology	Main Results	References
*ANRIL*	Aortic soy muscle cells/cardiovascular in vitro models	Coronary artery disease Type 2 diabetes Intercranial aneurisms	Associated with susceptibility to cardiovascular disorder	[114]
*BANCR*	Different melanoma cell lines mice xenografts	Melanoma	Associated with proliferation, malignancy and prognosis of melanoma via MAPK pathway	[115,116]
*FAS-AS1*	Lymphoma cell lines	Autoimmune diseases Lymphoma	Found downregulated in lymphoma cells is involved in sFAS regulation and alternative splicing	[117]
*HOXD-AS1*	Neuroblastoma cell lines	Neuroblastoma	Marker of neuroblastoma progression and malignancy	[118]
*HOTAIR*	Synoviocytes	Synovial inflammation Osteoarthritis	Downregulation reduces inflammation and osteoarthritis via Wnt/β-catenin pathway	[119]
*Lnc13*	Human intestinal sections and cells, macrophages	Celiac disease (CeD)	Associated with a wide panel of genes involved in inflammation in CeD	[120,121]
*Linc-Maf-4*	Patient macrophages	Multiple sclerosis	Associated with the pathogenesis and the severity of multiple sclerosis	[122]
*LncRNA-CMPK2*	Human cell lines	HCV infection	Correlated with HCV infection via	[123]
*lncRNADQ786243*	Chron’s disease patients	Chron’s disease	Correlated with Chron’s disease development via FOXP3	[16]
*LncRNA-p21*	Rheumatoid arthritis patients’ T cells	Rheumatoid arthritis	Upregulated in response to Metotrexate to control the pathology	[124]
*lncRNA RMPR*	Cartilage-hair hypoplasia patients	Cartilage-hair hypoplasia	Mutations associated with autoimmune pathologies in joints and kidneys and hematological abnormalities	[98,106,107,125]
*lncRNA SAS-ZFAT*	Autoimmune thyroid disease patients	Autoimmune thyroid disease(AITD)	Increases susceptibility to AITD	[126]
*MALAT1*	HREC cells	Diabetic retinopathy	Correlated to significant increases of his levels and associated inflammatory transcripts	[127]
*MORRBID*	Hypereosinophilic syndrome patients and myeloid cells	Hypereosinophilic syndrome	Potential therapeutic target for inflammatory disorders characterized by short-lived aberrant myeloid cell duration.	[82]
*NRIR*	Systemic sclerosis	Systemic sclerosis	Upregulated together with INF response in SSc patients	[82]
*OLA1P2*	Colorectal cancer patients	Colorectal patients	Correlated with the progression of the tumor and to poor prognosis	[128]

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
