# Peer review of "LncRNAs and Immunity: Coding the Immune System with Noncoding Oligonucleotides"

_ijms, 2021, doi:10.3390/ijms22041741_

Round 1
Reviewer 1 Report
In this paper, Bocchetti et al. review some of the long noncoding RNAs that have been involved in immune response pathways. The review harbors a lot of information but I feel it is difficult to follow. The sentences are sometimes too long and the different sections do not follow the same organization which makes the read complicated.
Specific comments:
- Authors mention that lncRNAs are 2 or more exons long, this should be corrected as single exon lncRNAs do also exist.
- In the inflammatory response section, they explain NFkB pathway in detail. They end up the section mentioning that NFkB together with JAK-STAT and MAPK pathway are the main inflammatory response drivers. Not sure why they do not introduce a bit the other two transcription factors too.
- A lot of lncRNAs are briefly explained, I suggest they are gathered in subgroups depending on their function and involvement in disease development.
General comments
- lncRNA and gene names have to be written in italics
- Grammar and sentence construction has to be revised.
Reviewer 2 Report
In the manuscript “LncRNAs and immunity: coding the immune system with non-2 coding oligonucleotides” the authors provide a very detailed review on lncRNAs related to immunity and associated with immune mediated diseases. The information is presented almost in the encyclopedic way. Each lncRNA is separately described based on currently available information. This makes the manuscript an excellent resource of knowledge but at the same time not very exciting. Shortening of these descriptions and providing more developed discussion/view on lncRNAs in immunity would probably make the paper more compelling.
Some minor issues:
- Some parts, like the one describing lncRNA functions (lines 163-197) or describing lncRNA and regulation of immunogenic expression (lines 198-218) are a little bit chaotic, the authors jump from one aspect to another. It seems like the authors tried to cover to many things in short sections. I would suggest some editing of these two parts of the manuscript to make them more organized.
- Gene names should be written in italic.
- Descriptions of tables are missing.
- Tables 1,2 and 3 – the width of columns should be adjusted to the text, columns with references could be narrowed so there will be more space for the “Main results” columns.
Author Response
In the manuscript “LncRNAs and immunity: coding the immune system with non-2 coding oligonucleotides” the authors provide a very detailed review on lncRNAs related to immunity and associated with immune mediated diseases. The information is presented almost in the encyclopedic way. Each lncRNA is separately described based on currently available information. This makes the manuscript an excellent resource of knowledge but at the same time not very exciting. Shortening of these descriptions and providing more developed discussion/view on lncRNAs in immunity would probably make the paper more compelling.
Thank you very much for your comments. In the conclusion we provided a new broader discussion about the reasons to push research on lncRNAs in innate and adaptive immunity response. Especially focusing on their importance as new potential biomarkers for the diagnosis of autoimmune diseases, but also to exploit their regulation as a new tool for precision medicine. The information presented in a schematic way was intended to make easier for the reader to find information in a simple and fast way, but at the same time we reduced some descriptions by making them more discursive.
Some minor issues:
1. Some parts, like the one describing lncRNA functions (lines 163-197) or describing lncRNA and regulation of immunogenic expression (lines 198-218) are a little bit chaotic, the authors jump from one aspect to another. It seems like the authors tried to cover to many things in short sections. I would suggest some editing of these two parts of the manuscript to make them more organized.
We divided the lncRNA paragraph in two sub paragraphs, one for molecular features and one for mechanisms. Moreover we removed or changed some long and unclear sentences as suggested.
2. Gene names should be written in italic.
Amended as suggested
3. Descriptions of tables are missing.
Tables description added at the bottom as suggested
4. Tables 1,2 and 3 – the width of columns should be adjusted to the text, columns with references could be narrowed so there will be more space for the “Main results” columns.
Tables were reformatted according to the clear reviewer suggestions.

Reviewer 3 Report
This review focuses on the lncRNA function in adaptive and innate immunity After a through introduction, the manuscript goes into specific details about individual lncRNAs and their role in innate and adaptive immunity as well as immune related diseases. As the authors pointed out, lncRNAs are involved in a number of functionally diverse physiological processes. This manuscript breaks down a number of individual links discusses the their individual contribution to immunity. I did find the inclusion of PAN to be a bit odd since this a a viral lncRNA and the manuscript was primarily focused on human lncRNAs. Many viruses encode lncRNAs so this many be a bit outside the focus of the manuscript. Overall, the manuscript is organized, well written, and provides a comprehensive overview of an important topic.
Author Response
Thank you very much for your comments. We removed PAN as suggested.
Major changes within the text are highlighted in red.

Round 2
Reviewer 1 Report
The paper has been considerably improved and I believe it can be published in its current form.